# Patterns of orthopedic and trauma admissions to a tertiary teaching and referral health facility in Kenya: Chart review

**Maxwell Philip Omondi**[1]*, **Joseph Chege Mwangi**[1], **Fred Chuma Sitati**[1], **Herbert Onga'ngo**[2]

**1** Department of Orthopedics, University of Nairobi, Nairobi, Kenya, **2** Department of Orthopedics, Kenyatta National Hospital, Nairobi, Kenya

* maxwellomondi@gmail.com

**Data Availability Statement:** All relevant data are within the paper and Supporting Information files.

**Funding:** MPO received funding from Kenyatta National Hospital for data collection and data

## Abstract

Tertiary hospitals in resource-limited countries should treat referred patients but in reality, are the first level of care for the vast majority of patients. As a result, the tertiary facility effectively functions as a primary health care facility. The urban phenomenon of widespread self-referral is associated with low rates of formal referral from peripheral health facilities. The study objective was to determine the patterns of orthopaedic and trauma admissions to Kenyatta National Hospital. This was descriptive study design. 905 patient charts were reviewed in 2021. The mean age was 33.8 years (SD 16.5) with range of 1–93 years. Majority 66.3% were between 25–64 years with those above 65 years being 40 (4.4%). Children 0–14 years comprised 10.9% of the admissions. Of the 905 admissions, 80.7% were accident and trauma-related admissions while 17.1% were non-trauma related admissions. About 50.1% were facility referrals while 49.9% were walk-ins. Majority of admissions were through Accident and Emergency Department 78.1%, Corporate Outpatient Care 14.9% and orthopedic Clinic 7.0%. About 78.7% were emergency admissions while 20.8% were elective admissions. Approximately 48.5% were due to Road Traffic Accidents and 20.9% due to falls. Close to 44.8% were casual workers and 20.2% unemployed. About 34.0% attained primary education and 35.0% secondary education. About 33.2% of female admissions were due to non-trauma conditions as compared to male admissions (12.8%) (p<0.001). Admissions for those aged 25–64 years were 3.5 more likely to have emergency admission as compared to those aged 0–14 years. Male were 65.1% less likely to have elective admissions compared to female (p<0.001). Whereas lower limb injuries and non-trauma related conditions were the most commonly admitted conditions, Lower limb injury and spine cases were mostly facility referred while non-trauma conditions were walk-in patients. Vast majority (89.2%) of admissions were from Nairobi Metropolitan region.

## Introduction

Approximately 90% of the estimated traumatic injuries occur in low and middle-income countries according to World Health Organization (WHO) [1] and this represents an important

analysis. The rest of the authors received no funding for this work. The funders had no role in study design, data collection and analysis, decision to publish, or preparation of the manuscript.

**Competing interests:** The authors have declared that no competing interests exist.

global public health problem now and in the coming years [2, 3]. Admission patterns for trauma cases have shown gender bias with men affected more than females. The young population age-group less than 40 years are affected as compared to the older population with the most common injury mechanism being road traffic accidents, falls, assaults.

Tertiary hospitals in resource-limited countries should treat referred patients but in most cases are the first level of care for the vast majority of patients [4]. One of the challenges in health care delivery in resource-limited settings is inappropriate utilization of tertiary health facilities that results in patients' congestion in referral hospitals with simple conditions that can be effectively managed at the lower peripheral health facilities. The majority of these patients are self-referred, bypassing lower-level health facilities in the process [5–8]. The reason for bypassing nearest health facilities seems to be multifactorial. Factors such as patients' perception of high quality of health care and resource availability at referral hospitals play a role [6, 9]. A study done in Lusaka demonstrated how University Teaching Hospital is bypassed and as a result the tertiary facility effectively functions as a primary health care facility. The urban phenomenon of widespread self-referral is associated with low rates of formal referral from peripheral health facilities [7, 10]. The lower limbs did get injured more often than the upper limbs, spine, and pelvis. However, the paucity of data regarding the patterns of admissions in developing countries and particularly sub-Saharan Africa. The few studies done in the Western context focuses on traumatic orthopaedic injuries and the non-traumatic orthopaedic cases have been left unaddressed. Understanding the patterns of admission is important for patient planning and management and in a university facility, this is critical in planning for quality training.

KNH was established as a National Referral and Teaching Hospital, to provide training and medical research. KNH was established in 1901 and became a State Corporation in 1987 and sits at the peak of the health referral system in Kenya [11]. According to the KNH Board order of 1987 contained in the Legal Notice No. 109, the functions of KNH were spelled out as a) to receive patients on referral for specialized health care; b) to provide facilities for medical education for the University of Nairobi and other health allied courses; c) to contribute to national health planning [11]. This understanding has been reinforced by the Kenya Health Sector Referral Implementation Guidelines, 2014, and the Constitution of Kenya 2010 which tasks KNH with the responsibility for health policy formulation [12–14]. KNH has over 6000 personnel with a bed capacity of 2000 with several specialized units well-resourced with highly qualified personnel and equipment's namely internal medicine, orthopedic and trauma surgery, pediatric surgery, plastic and reconstruction surgery, general surgery, neurosurgery, cardiothoracic and vascular surgery, laboratory, radiology and imaging, renal, critical care units for adults and pediatrics.

However, there is paucity of data regarding the patterns of orthopaedic and trauma admissions in developing countries and particularly sub-Saharan Africa. The purpose of this study is to determine patterns of orthopaedic and trauma admissions to KNH.

## Materials and methods

### Ethical statement

UoN/KNH Ethics and Research Committee granted ethical approval (ERC No: P852/10/2021). Administrative approval was also sought from KNH Medical Research Department and KNH Orthopaedics Department.

### Study design

This was a descriptive study design.

## Study area

KNH is located in Nairobi, Kenyans Capital City, along the Hospital Road in Upper Hill– 5km from the city center. On the northern side, the hospital faces Ngong Road near its roundabout with Mbagathi Road [15]. On the eastern side is Hospital Road to the West is Mbagathi Road and to the south of the hospital compound is the Nairobi-Kisumu railway line. KNH Ortho-paedic Wards are housed on the 6th floor.

## Study period

1st February to 31st December 2021.

## Study population

Orthopaedic and trauma inpatient caseload.

## Inclusion criteria

All orthopaedic and trauma admissions to KNH during the study period.

## Sample size

A total 0f 905 charts were reviewed.

## Sampling procedure

Stratified sampling technique was used. Strata was based on the point of admission: Accident and Emergency (A&E), Orthopaedic Clinic (OC) and Corporate Outpatient Centre (COC). The Population Proportion to Size (PPS) was then used to determine the sample for each stra-tum based on monthly admissions (Table 1). Within each strata systematic sampling was used to sample the charts for review based on the total monthly admissions per point of admission.

## Recruitment and consenting procedures

The orthopedic and trauma admissions were identified from the admission desk of Health Information System at a) Accident and Emergency Unit b) Orthopedic clinic c) Corporate Outpatient Center.

**Table 1. Orthopaedic and trauma admissions to KNH stratified by point of admission, 2021.**

| Month of the year, 2021' | Point of admission | | | |
|---|---|---|---|---|
| | A&E | OC | COC | Total |
| February | 94 | 10 | 9 | 113 |
| March | 68 | 4 | 9 | 81 |
| April | 79 | 3 | 10 | 92 |
| May | 67 | 3 | 11 | 81 |
| June | 78 | 5 | 9 | 92 |
| August | 62 | 10 | 15 | 87 |
| September | 66 | 8 | 14 | 88 |
| October | 82 | 8 | 12 | 102 |
| November | 45 | 6 | 27 | 78 |
| December | 66 | 6 | 19 | 91 |
| Total | 707 | 63 | 135 | 905 |

## Variables

Age, sex, marital status, religion, Sub- County of residence, County of residence, country of residence, education level, Nature of admission, Occupation, Type of admission, Diagnoses, Mode of payment, Mechanisms of injury. All these variables were extracted from patients charts in the health records office.

## Quality assurance & quality control procedures

A pilot study was conducted during the design of the study protocol to test the data collection tools for relevance, appropriateness to answer the research questions and adjustments of the data collection tools made as necessary. Daily reviews of all the abstracted forms were conducted and verified for accuracy, completeness, and compliance to the research protocol.

## Data management, analysis

The data were entered into a password-protected Redcap database kept by the KNH Medical Research Department. The quantitative data was analyzed using SPSS version 21. Patterns of orthopedic and trauma admissions were determined using descriptive statistics such as frequencies, measures of central tendencies, measures of dispersions while inferential statistics will be calculated using Pearson's chi-squared tests and Logistic regressions.

## Study limitations

Missing and incomplete data–this was be minimized by making phone calls to patients and or relatives/guardians to fill in the missing information. The patients and relatives/guardians contacts were retrieved from patients' charts.

## Results

### Basic profile of the sample population

The overall mean age was 33.8 years (SD 16.5) with range of 1–93 years. Majority 600 (66.3%) were between 25–64 years with those above 65 years being 40 (4.4%). Children 0–14 years comprised 99 (10.9%) of the orthopaedic and traumatic admissions. About 743 (82.7%) were Accident/trauma-related admissions while 155 (17.3%) were non-trauma related admissions. About 453 (50.1%) were facility referrals while 452 (49.9%) were walk-ins. With regard to the type of admission 712 (79.1%) were emergency admissions while 188 (20.9%) were elective admissions. With respect to the mechanism of injury, 439 (48.7%) were due to Road Traffic Accidents, 189 (21.0%) were due to falls and non-trauma related conditions 155 (17.2%) (Table 2).

About 703 (77.7%) were male and 198 (21.9%) were female. About 405 (45.3%) were casual workers, 183 (20.5%) unemployed. Education level was also reviewed with 308 (34.8%) having primary education and 317 (35.9%) having secondary education (Table 2).

Majority of admissions were through A&E regardless of the age category with significant admissions for those above 65 years of age being admitted through COC (p<0.001) (Table 3). While it was noted that a higher proportion of male 574 (81.9%) and females (127 (64.1%) were admitted through A&E, proportionately more females were admitted through COC (26.3%) (p<0.001) (Table 3). Admissions for those with secondary level of education and below were majorly admitted through the A& E. For those with tertiary level of education, a significant proportion were admitted through the COC (41.8%) (p<0.0001) (Table 3).

There was no statistically significant association between age, sex, occupation, education and the nature of admission (p>0.05) (Table 4).

**Table 2. Basic profile of the sample population (N = 905), February–December 2021.**

| Variable | Category | Frequency n (%); 95% CI | Variable | Category | Frequency n (%);95% CI |
|---|---|---|---|---|---|
| Age | 0–14 years | 99 (10.9%;9.0% - 13.2%) | Occupation | Businessman/woman | 112 (12.5%;10.4% -14.9%) |
| | 15–24 years | 166 (18.3%; 15.9% - 21.0%) | | Casual | 405 (45.3%;42.0% - 48.6%) |
| | 25–64 years | 600 (66.3%;63.1% - 69.4%) | | Employed | 135 (15.1%;12.8% - 17.6%) |
| | Above 65 years | 40 (4.4%; 3.2% - 6.0%) | | Other | 59 (6.6%;5.1% - 8.4%) |
| Sex | Female | 198 (21.9%;19.3% - 24.8%) | | Unemployed | 183 (20.5%;17.87% - 23.3%) |
| | Male | 703 (77.7%;75.2% - 80.7%) | | Missing | 11 |
| | Missing | 4 | Education | None | 55 (6.2%;4.7% - 8.0%) |
| Marital status | Married | 446 (49.3%;46.0% -52.7%)) | | Pre-school | 22 (2.5%;1.6% - 3.7%) |
| | Minor | 110 (12.2%;10.1% - 14.5%) | | Primary | 308 (34.8%;31.7% - 38.1%) |
| | Separated & divorced | 49 (5.4%;4.1% - 7.1%) | | Secondary | 317 (35.9%;32.7% - 39.1%) |
| | Single | 275 (30.4%;27.4% - 33.5%) | | Tertiary | 182 (20.6%;18.0% - 23.4%) |
| | Widow | 24 (2.7%;1.7% - 3.9%) | | Missing | 21 |
| | Missing | 1 | Point of admission | A& E | 707 (78.1%;75.3% - 80.8%) |
| Religion | Atheist | 3 (0.3%;0.03% - 0.09%) | | Clinic | 63 (7.0%;5.4% - 8.8%) |
| | Christian | 865 (96.65%;95.3% - 97.7%) | | COC | 135 (14.9%;12.7% - 17.4%) |
| | Hindu | 3 (0.3%;0.03% -0.09%) | Smoking | No | 680 (79.3%;76.4% - 81.9%) |
| | Muslim | 24 (2.7%;1.7%-4.0%) | | Yes | 178 (20.8%;18.1% - 23.6%) |
| | Missing | 10 | | Missing | 47 |
| Nature of Injury | Accident/trauma | 743 (82.7%;80.1% - 85.2%) | Alcohol | No | 515 (60.0%;56.6% - 63.3%) |
| | Non-trauma | 155 (17.3%;14.8%-19.9%) | | Yes | 344 (40.0%;36.8% - 43.4%) |
| | Missing | 7 | | Missing | 46 |
| Mode of payment | Cash | 621 (68.9%;65.7%-71.9%) | Type of admission | Elective | 188 (20.9%;18.3% - 23.7%) |
| | Insurance | 281 (31.1%;28.1%-34.3%) | | Emergency | 712 (79.1%;76.3% - 81.7%) |
| | Missing | 3 | | Missing | 5 |
| Mechanism of Injury | Assault | 31 (3.4%;2.4%-4.9%) | Nature of admission | Walk-in | 452 (49.9%; 46.6% - 53.3%) |
| | Fall | 189 (21.0%;18.4%-23.8%) | | Facility referral | 453 (50.1%;46.8% - 53.4%) |
| | Gunshot | 6 (0.7%;0.2% -1.4%) | *Referrals | With referral letter | 219 (48.7%;44.0% - 53.4%) |
| | Non-Trauma | 155 (17.2%;14.8% - 19.8%) | | Without referral letter | 231 (51.3%;46.6% - 56.0%) |
| | Others | 81 (9.0%;7.2% - 11.1%) | | Missing | 3 |
| | RTA | 439 (48.7%;45.4%- 52.0%) | | | |
| | Missing | 4 | | | |

*Referrals were 453 but 3 charts did not indicate if they were written referral letter or not. Missing variables varied from 1 to 47 but they were excluded from proportional analysis for the variables concerned. A 95% CI was computed for each of the point estimates.

A significant proportion of those aged 0–14 years (44.9%) and above 65 years (37.5%) were due to falls whereas those between 15–24 years (62.7%) and 25–64 years (51.4%) were due to RTA (p<0.001) (Table 5. Significant proportion of female admissions (33.2%) were due to non-trauma conditions as compared to male admissions (12.8%) (p<0.001) (Table 5).

Admissions for those aged 25–64 years were 70.9% less likely to have elective admissions as compared to those aged 0–14 years (Table 6).

Male sex was 2.860 (2.009–4.070) more likely to have emergency admissions compared to female while those employed were 74.1% less likely to have emergency admissions compared to businessmen/women (Table 6). Patients with some levels of education were less likely to have emergency admissions compared with those no education (Table 6).

The study revealed no statistically significant association between the different age categories and mode of payment (Table 7). Male admissions were 2.003 (1.445–2.775) more likely to

**Table 3. Association between key socio-demographic characteristics and point of admission.**

| Variable | Category | Point of admission | | | Chi-square; p-value |
|---|---|---|---|---|---|
| | | A&E | OC | COC | |
| Age | 0–14 years | 72 (72.7%) | 15 (15.2%) | 12 (12.1%) | 27.938; p<0.001 |
| | 15–24 years | 147 (88.6%) | 7 (4.2%) | 12 (7.2%) | |
| | 25–64 years | 460 (76.9%) | 38 (6.4%) | 100 (16.7%) | |
| | Above 65 years | 26 (65.0%) | 3 (7.5%) | 11 (27.5%) | |
| Sex | Female | 127 (64.1%) | 19 (9.6%) | 52 (26.3%) | 30.045; p<0.001 |
| | Male | 574 (81.9%) | 44 (6.3%) | 83 (11.8%) | |
| Education | None | 44 (80.0%) | 5 (9.1%) | 6 (10.9%) | 138.593; p<0.001 |
| | Pre-school | 21 (95.5%) | 1 (4.5%) | 0 (0.0%) | |
| | Primary | 262 (85.3%) | 23 (7.5%) | 22 (7.2%) | |
| | Secondary | 264 (83.5%) | 27 (8.5%) | 25 (7.9%) | |
| | Tertiary | 100 (54.9%) | 6 (3.3%) | 76 (41.8%) | |

Pearsons's chi-square test was used to test for association at 5% level of significance. A&E, OC and COC.

be cash payers as compared to female (Table 7). This means that female admissions tended to have active insurance cover compared to male admissions.

With regard to occupation status, those who were unemployed were 70.4% less likely to be cash payers as compared to businessmen/women. Admissions who were smokers were 2.097 (1.402–3.136) more likely to be cash payers as compared to those who did not smoke. In addition, admissions who took alcohol were 2.134 (1.562–2.917) more likely to be cash payers as compared to those who did not take alcohol (Table 7).

Lower limb injuries and non-trauma related conditions were the most commonly admitted conditions. Amongst non-trauma conditions, vast majority were walk-in patients who were

**Table 4. Association between key socio-demographic characteristics and nature of admission.**

| Variable | Category | Nature of admission | | Chi-square, p-value |
|---|---|---|---|---|
| | | Walk-in | Facility Referral | |
| Age | 0–14 years | 38 (38.4%) | 61 (61.6%) | 6.295; p = 0.098 |
| | 15–24 years | 87 (52.4%) | 79 (47.6%) | |
| | 25–64 years | 305 (50.8%) | 295 (49.2%) | |
| | Above 65 years | 22 (55.0%) | 18 (45.0%) | |
| Sex | Female | 111 (56.1%) | 87 (43.9%) | 0.056; p = 0.056 |
| | Male | 340 (48.4%) | 363 (51.6%) | |
| Occupation | Businessman/woman | 58 (51.8%) | 54 (48.2%) | 1.754; p = 0.781 |
| | Casual | 200 (49.4%) | 205 (50.6%) | |
| | Employed | 72 (53.3%) | 63 (46.7%) | |
| | Other | 26 (44.1%) | 33 (55.9%) | |
| | Unemployed | 89 (48.6%) | 94 (51.4%) | |
| Education | None | 28 (50.9%) | 27 (49.1%) | 6.886; p = 0.142 |
| | Pre-school | 7 (31.8%) | 15 (68.2%) | |
| | Primary | 142 (46.1%) | 166 (53.9%) | |
| | Secondary | 159 (50.2%) | 158 (49.8%) | |
| | Tertiary | 101 (55.5%) | 81 (44.5%) | |

Pearsons's chi-square test was used to test for association at 5% level of significance.

**Table 5. Association between the socio-demographic characteristics and the mechanism of injury.**

| Variable | Category | Mechanism of injury | | | | | | Chi-square;p-value |
|---|---|---|---|---|---|---|---|---|
| | | Assault | fall | gunshot | Non-trauma | Others | RTA | |
| Age | 0–14 years | 1 (1.0%) | 44 (44.9%) | 0 (0.0%) | 18 (18.4%) | 15 (15.3%) | 20 (20.4%) | 91.598; p<0.001 |
| | 15–24 years | 6 (3.6%) | 22 (13.3%) | 2 (1.2%) | 16 (9.6%) | 16 (9.6%) | 104 (62.7%) | |
| | 25–64 years | 24 (4.0%) | 108 (18.1%) | 4 (0.7%) | 108 (18.1%) | 46 (7.7%) | 307 (51.4%) | |
| | Above 65 years | 0 (0.0%) | 15 (37.5%) | 0 (0.0%) | 13 (32.5%) | 4 (10.0%) | 8 (20.0%) | |
| Sex | Female | 1 (0.5%) | 49 (25.0%) | 0 (0.0%) | 65 (33.2%) | 16 (8.2%) | 65 (33.2%) | 59.162; p<0.001 |
| | Male | 30 (4.3%) | 139 (19.8%) | 6 (0.9%) | 90 (12.8%) | 64 (9.1%) | 372 (53.1%) | |

Pearsons's chi-square test was used to test for association at 5% level of significance.

self-referred. Lower limb injury (p<0.008) and spine cases were mostly facility referred (p<0.001). While the bulk of acetabular and pelvic injuries were facility referred but these were not statistically significant (Table 8).

Most of the admissions were from Nairobi County at 60.3% with 89.2% of patients coming within Nairobi Metropolitan area (a radius of about 40km from Nairobi) Fig 1.

## Discussions

The study revealed the mean age for admissions were 33.8 years with majority between 25–64 years. Children and those above 65 years of age were the minority age groups. This compares with studies done in Rwanda, Uganda, Botswana, India, United States of America, Brazil that showed orthopaedic and trauma admissions are of younger age group [16–22].

It was noted that a higher proportion of admissions were through A&E and this is in tandem with the finding that about three-fourth of the orthopaedic and trauma admissions were emergency. However, a higher proportion of females were admitted through COC which is a private wing of KNH and these tended to be older women above 65 years of age with fragility fractures and musculoskeletal degenerative disorders. These were women with active insurance cover and were presumably of higher economic status.

**Table 6. Association between key socio-demographic characteristics and type of admission.**

| Variable | Category | Emergency | Elective | | Logistic regression; OR (95% CI) |
|---|---|---|---|---|---|
| Age | 0–14 years | 72 (73.5%) | 26 (26.5%) | 13.550; p = 0.004 | 1.0 |
| | 15–24 years | 147 (88.6%) | 19 (11.4%) | | 0.812 (0.360–1.835) |
| | 25–64 years | 466 (78.1%) | 131 (21.9%) | | *0.291 (0.127–0.668)* |
| | Above 65 years | 27 (69.2%) | 12 (30.8%) | | 0.633 (0.312–1.283) |
| Sex | Female | 125 (63.8%) | 71 (36.2%) | 35.813; p<0.001 | 1.0 |
| | Male | 584 (83.4%) | 116 (16.6%) | | *2.860 (2.009–4.070)* |
| Occupation | Businessman/woman | 82 (73.2%) | 30 (26.8%) | 66.366; p<0.001 | 1.0 |
| | Casual | 361 (89.8%) | 41 (10.2%) | | 0.833 (0.416–1.669) |
| | Employed | 80 (59.3%) | 55 (40.7%) | | *0.259 (0.136–0.491)* |
| | Other | 41 (69.5%) | 18 (30.5%) | | 0.626 (0.324–1.208) |
| | Unemployed | 142 (78.5%) | 39 (21.5%) | | 1.566 (0.816–3.005) |
| Education | None | 46 (83.6%) | 9 (16.4%) | 55.091; p<0.001 | 1.0 |
| | Pre-school | 20 (90.9%) | 2 (9.1%) | | *0.292 (0.135–0.633)* |
| | Primary | 260 (85.2%) | 45 (14.8%) | | *0.149 (0.034–0.658)* |
| | Secondary | 264 (83.8%) | 51 (16.2%) | | *0.258 (0.168–0.399)* |
| | Tertiary | 109 (59.9%) | 73 (40.1%) | | *0.288 (0.189–0.440)* |

**Table 7. Association between key socio-demographic characteristics and mode of payment.**

| Variable | Category | Mode of payment | | Chi-square test | Logistic regression; OR (95% CI) |
|---|---|---|---|---|---|
| | | Cash | Insurance | | |
| Age | 0–14 years | 57 (57.6%) | 42 (42.4%) | 9.790; p = 0.020 | 1.0 |
| | 15–24 years | 117 (71.3%) | 47 (28.7%) | | 0.997 (0.474–2.095) |
| | 25–64 years | 423 (70.8%) | 175 (29.2%) | | 0.543 (0.267–1.108) |
| | Above 65 years | 23 (57.5%) | 17 (42.5%) | | 0.558 (0.291–1.071) |
| Sex | Female | 112 (56.6%) | 86 (43.4%) | 17.775; p<0.001 | 1.0 |
| | Male | 506 (72.0%) | 194 (27.6%) | | *2.003 (1.445–2.775)* |
| Occupation | Businessman/woman | 82 (73.9%) | 29 (26.1%) | 97.262; p<0.001 | 1.0 |
| | Casual | 331 (82.1%) | 72 (17.9%) | | 0.481 (0.247–0.938) |
| | Employed | 53 (39.3%) | 82 (60.7%) | | 0.842 (0.464–1.529) |
| | Other | 34 (57.6%) | 25 (42.4%) | | *2.104 (1.130–3.916)* |
| | unemployed | 113 (61.7%) | 70 (38.3%) | | *0.296 (0.166–0.526)* |
| Education | None | 45 (81.8%) | 10 (18.2%) | 56.214; p<0.001 | 1.0 |
| | Pre-school | 17 (77.3%) | 5 (22.7%) | | 0.197 (0.093–0.414) |
| | Primary | 237 (77.2%) | 70 (22.8%) | | 0.260 (0.092–0.736) |
| | Secondary | 224 (70.9%) | 92 (29.1%) | | 0.262 (0.176–0.388) |
| | Tertiary | 85 (47.0%) | 96 (53.0%) | | 0.364 (0.249–0.532) |
| Smoking | No | 447 (65.9%) | 231 (34.1%) | 13.386; p<0.001 | 1.0 |
| | Yes | 142 (80.2%) | 35 (19.8%) | | 2.097 (1.402–3.136) |
| Alcohol | No | 321 (62.7%) | 191 (37.3%) | 23.087; p<0.001 | 1.0 |
| | Yes | 269 (78.2%) | 75 (21.8%) | | 2.134 (1.562–2.917) |

Pearsons's chi-square test and Logistic regression was used to test for association at 5% level of significance.

Admissions for those with secondary level of education and below were majorly admitted through the A& E since these were mostly casual workers caught up in road traffic accidents. For those with tertiary level of education, a significant proportion were admitted through the COC and most of these patients had active insurance cover, better educated and presumably of higher socio-economic status.

About two-thirds of the admissions were either casuals or unemployed. This compares favourably with study done in Taiwan that revealed orthopaedic fractures were associated with patients of low socio-economic status [23]. However, it contrasted with studies done on orthopedic admissions in Kilimanjaro Christian Medical Centre (KCMC) in Northern Tanzania

**Table 8. Type of orthopedic and trauma admissions stratified by nature of admission.**

| Type of injury | Walk-in | Facility referral | p-value |
|---|---|---|---|
| Spine injury | 19 (4.9%) | 48 (12.2%) | *<**0.001*** |
| Acetabular injury | 5 (1.3%) | 9 (2.3%) | 0.283 |
| Pelvic injury | 17 (4.4%) | 27 (6.9%) | 0.124 |
| Lower limb injury | 156 (40%) | 195 (49.6%) | ***0.008*** |
| Upper limb injury | 51 (13.1%) | 71 (18.1%) | 0.053 |
| Non -Trauma conditions | 142 (36.4%) | 43 (10.9%) | *<**0.001*** |

Pearsons's chi-square test was used to test for association at 5% level of significance. Non-trauma conditions refer to orthopaedic conditions such as osteomyelitis, implant removal, ganglion cysts, soft tissue infections of the extremities, carpal tunnel syndrome amongst others.

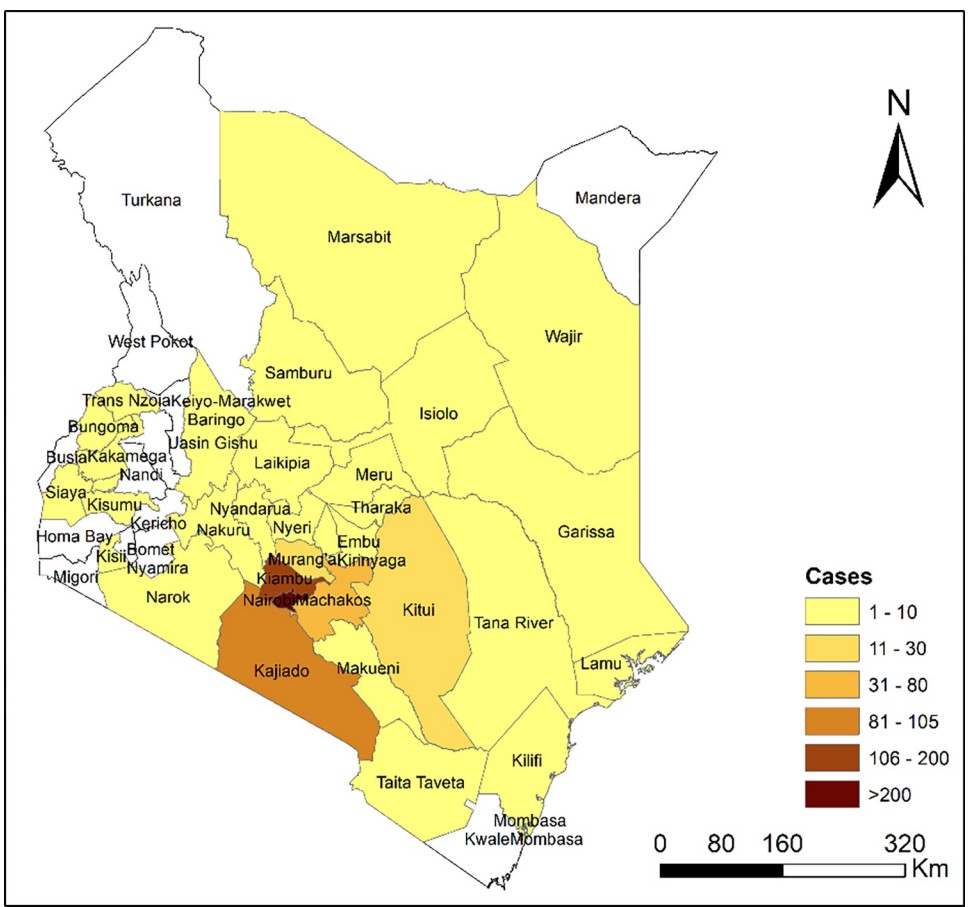

**Fig 1. Spatial distribution of orthopedic and trauma admissions to KNH, February–December 2021.** Republished from [https://www.diva-gis.org/datadown] under a CC BY license, with permission from [Robert J. Hijmans], original copyright [2017]. The map was produced in QGIS version 3.24 (QGIS Development Team 2014). The data representing the administrative units were obtained from the DIVA-GIS website (https://www.diva-gis.org/datadown) in shapefile format.

showed the three most common occupations were farmers, businessman, professional drivers [24, 25]. This could be because KCMC is a private health facility as opposed to KNH which is a public health facility. Those who were unemployed were also less likely to have insurance cover compared to businessmen and women and this could be attributed to their low-socio-economic status. Furthermore, the study revealed that about a third of the admissions had active insurance cover. This contrasts with a multicenter observation study done on distribution of orthopedic fractures in low and middle-income countries revealed about 18% of orthopedic admissions in Africa had health insurance cover [26]. It also contrasts with a retrospective study done in PCEA Kikuyu Mission Hospital in Kenya showed about 60.82% of orthopedic patients have insurance cover [27]. This could be explained by the fact that PCEA Kikuyu Mission Hospital is a private health facility that admits patients with higher socio-economic status compared to KNH which is a public health facility. As a result, a number of patients admitted are unable to pay for the required orthopedic implants, orthosis and even paying for their medical bills upon discharge and thus contributing to congestion in the wards due to delay in treatment and increased length of hospital stay.

Over three-quarters of admissions were males. This compares favourably with studies done in Taiwan, India, Botswana, Tanzania, Rwanda, South Africa and Brazil that reveals male

predominance [16, 19–21, 23, 28–30]. However, this contradicts a retrospective study done in England that revealed approximately equal male to female ratio [28]. A significant proportion of female admissions were due to non-trauma conditions as compared to male admissions and this could be due to an increase in female admissions above 65 years of age due to musculo-skeletal degenerative disorders. In addition, male admissions predominated and peaked at 25–64 years and these admissions declined steadily to 65years of age and admissions were comparable across gender for those above 65 years of age. This compares with a retrospective study done in a tertiary Hospital in Nepal that showed similar admission rates from 60 years of age [31]. Male were less likely to have elective admissions compared to female and this is explained by the fact that majority were emergency admissions which males predominated.

About half of admissions were facility referrals while the other half were walk-ins. Of the facility referrals about half had written referral letters. This contradicts study done in Meru, Kenya that showed self-referrals accounted for about 80% of referrals to Meru District Hospital [32]. This could be due to the fact that the study focussed on non-surgical patients. A similar study done in Kwazulu Natal, South Africa revealed that self-referrals were 36% with 64% being appropriately referred with written referral letter [33]. On the other hand a study done in Nigeria showed 92.9% of patients in tertiary hospital directly went to the facility without a referral letters while 7.1% were facility referred [34]. The disparities of these findings could be due to the fact that this study specifically reviewed orthopaedic and trauma admissions and not all medical conditions. Nevertheless, the fact that half of the admissions are walk-ins means that self-referral to KNH is rampant and unnecessarily simple conditions that should have been managed at lower tier health facilities, end up being managed at KNH. This strains the limited resources available at KNH that results in lower quality of care. It also has an effect of leading to underutilization of the lower-tier health facilities. For KNH to be able to perform its statutory obligations effectively as a premier referral health facility, then this proportion of walk-ins should be reduced through stringent enactment and enforcement of referral guidelines and also capacity building the lower-tier health facilities provide quality and timely essential orthopaedic and trauma care.

About half of admissions were due to RTA followed by falls and non-trauma related conditions. This compares favourably with studies done in former Rift Valley Provincial Hospital in Kenya, Tanzania, Rwanda, India, USA and Taiwan that revealed RTA was the major cause of orthopaedic trauma admissions [16, 17, 23–25, 30, 35, 36]. However, this differs with retrospective study done in South Africa that revealed interpersonal violence was the major mechanism of injury of orthopaedic admissions at 60% followed by RTA at 19% [21]. Also, a retrospective study done in Botswana and Tanzania showed falls as a leading cause of orthopaedic injuries and RTA second at 36% [29, 36]. Therefore, to reduce the burden of trauma management at KNH and the country at large there is need to control road carnage through enforcement of traffic laws. In this study falls were the commonest mechanism of injury amongst orthopaedic admissions in children and elderly population. This compares favourably with similar studies done in Middle East region, Botswana that showed falls were commonest cause of admissions for the extremes of age–children and the elderly [36, 37]. This also compares with a multicenter observation study done on distribution of orthopedic fractures in low and middle-income countries revealed about falls (64.5%) was the main mechanism of injury for those 60 years and above [26]. This can be explained by the understanding that elderly patients who are more prone to fragility fractures and children are more prone to falls.

However, there was no significant difference when it came to facility referrals being accompanied by official written referral letters from the referring facilities to KNH. This is because most of the referrals were verbal over the telephone and once a verbal consensus has been reached the referring health facilities did not see the need of writing an official referral letter.

The fact that lower limb injuries and non-trauma admissions form the bulk of walk-in could be due to patients' perception of better quality of service at KNH in otherwise conditions that can be treated at lower-level health facilities. Spine injuries were mainly facility referred and this reflects lack of spine surgeons and other specialized surgical expertise like plastic and reconstructive surgeons, neurosurgeons as well as implants and equipment's necessary to handle spine patients and they end up being referred to KNH for specialized care.

Vast majority of patients admitted came from within the Nairobi Metropolitan area consisting of Nairobi, Kajiado, Machakos and Kiambu counties. This compares with a review of orthopedic admissions in KCMC in Northern Tanzania showed 65.7% of the patients came from state of Kilimanjaro where the hospital is located, 12.7% from Arusha,6.4% from Tanga, 5.9% from Manyara and 1.5% from Singida [24]. A similar study done in Muhimbili National Hospital in Tanzania showed only 0.8% of admissions were from outside Dar Es Salaam [38]. A similar study done in Blantyre in Malawi also revealed majority of referrals come from within the Tertiary facility [39]. This reinforces the need to capacity build the referring health facilities within Nairobi Metropolitan region to be able to handle uncomplicated orthopedic and trauma conditions.

## Conclusions

In conclusion, majority of orthopaedic and trauma admissions were younger patients and mostly males. Women tended to have more admissions above 65 years of age mostly through COC. About equal proportions of admissions were facility referrals and walk-ins. About half the facility referrals had written formal referral letters. Majority of admissions were due to RTA followed by falls and non-trauma related conditions. About three-fourth of admissions were emergencies. Those with no education level were mostly associated with emergency admissions compared with those with some level of education. Those employed tended to be elective admissions. About a third of the admissions had active insurance cover. About two-thirds of the admissions were either casuals or unemployed. Lower limb injury and non-trauma related conditions were the most commonly admitted conditions. Lower limb injury and spine cases were mostly facility referred while non-trauma conditions were mostly walk-in patients. Vast majority of admissions were from within Nairobi Metropolitan region.

## Recommendations

1. Nairobi County government should focus on improving the human, infrastructure capacity and resource availability to government health facilities within Nairobi County to manage non-traumatic orthopaedic conditions and provide essential orthopaedic and trauma care to decongest KNH;

2. There is need to upscale the training of spine, pelvic and acetabular surgeons and provide necessary equipment's and implants to key referring health facilities to limit on referrals to KNH;

3. There is need to educate the general public on the role of KNH as a referral health facility and patients encouraged to visit lower-tier health facilities for essential orthopaedic and trauma care;

4. Given that RTAs are the dominant mechanism of injury the National Traffic Safety Authority (NTSA) should strive to enforcement traffic regulations in a bid to curb the carnage on our roads and these will reduce the orthopaedic and trauma caseloads in our health facilities;

5. There is need to set-up a Trauma registry at KNH to monitor the trends of trauma case-loads with a view to improve patient care as well as feed to policy makers for corrective action;

## Supporting information

**S1 Checklist. STROBE statement—Checklist of items that should be included in reports of *cross-sectional studies.***
(DOC)

**S1 Data. SPSS.**
(SAV)

## Author Contributions

**Conceptualization:** Maxwell Philip Omondi, Fred Chuma Sitati, Herbert Onga'ngo.

**Data curation:** Maxwell Philip Omondi, Herbert Onga'ngo.

**Formal analysis:** Maxwell Philip Omondi, Fred Chuma Sitati, Herbert Onga'ngo.

**Funding acquisition:** Maxwell Philip Omondi, Joseph Chege Mwangi.

**Investigation:** Maxwell Philip Omondi, Joseph Chege Mwangi.

**Methodology:** Maxwell Philip Omondi, Joseph Chege Mwangi, Fred Chuma Sitati.

**Project administration:** Maxwell Philip Omondi.

**Resources:** Maxwell Philip Omondi.

**Software:** Maxwell Philip Omondi.

**Supervision:** Maxwell Philip Omondi, Joseph Chege Mwangi, Fred Chuma Sitati, Herbert Onga'ngo.

**Validation:** Maxwell Philip Omondi, Joseph Chege Mwangi, Fred Chuma Sitati.

**Visualization:** Maxwell Philip Omondi, Joseph Chege Mwangi, Fred Chuma Sitati, Herbert Onga'ngo.

**Writing – original draft:** Maxwell Philip Omondi, Joseph Chege Mwangi, Fred Chuma Sitati, Herbert Onga'ngo.

**Writing – review & editing:** Maxwell Philip Omondi, Joseph Chege Mwangi, Fred Chuma Sitati, Herbert Onga'ngo.

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
