## [Decision Letter · Decision Letter 0]

6 Mar 2023

PGPH-D-22-01913

Patterns of orthopedic and trauma admissions to a tertiary teaching and referral health facility in Kenya: Chart review

Dear Dr. Omondi,

Thank you for submitting your manuscript to PLOS Global Public Health. After careful consideration, we feel that it has merit but does not fully meet PLOS Global Public Health’s publication criteria as it currently stands. Therefore, we invite you to submit a revised version of the manuscript that addresses the points raised during the review process.

The manuscript has been evaluated by four reviewers, and their comments are available below.

The reviewers have raised a number of concerns that need attention. They request additional contextual details, and suggest that you rewrite the manuscript to be more concise. 

Could you please revise the manuscript to carefully address the all concerns raised below?

We look forward to receiving your revised manuscript.

Kind regards,

Steve Zimmerman, PhD

PLOS Staff Editor

Journal Requirements:

1. Please send a completed 'Competing Interests' statement, including any COIs declared by your co-authors. If you have no competing interests to declare, please state "The authors have declared that no competing interests exist". Otherwise please declare all competing interests beginning with the statement "I have read the journal's policy and the authors of this manuscript have the following competing interests:"

b. If any authors received a salary from any of your funders, please state which authors and which funders.

3. Please provide separate figure files in .tif or .eps format only and remove any figures embedded in your manuscript file. Please also ensure that all files are under our size limit of 10MB.

4. In the online submission form, you indicated that "We have full access of the original raw data and will share the data on request on short notice". All PLOS journals now require all data underlying the findings described in their manuscript to be freely available to other researchers, either 1. In a public repository, 2. Within the manuscript itself, or 3. Uploaded as supplementary information.

5. Fig 1: please (a) provide a direct link to the base layer of the map (i.e., the country or region border shape) and ensure this is also included in the figure legend; and (b) provide a link to the terms of use / license information for the base layer image or shapefile. We cannot publish proprietary or copyrighted maps (e.g. Google Maps, Mapquest) and the terms of use for your map base layer must be compatible with our CC-BY 4.0 license. 

Additional Editor Comments (if provided):

Reviewers' comments:

Reviewer's Responses to Questions

**Comments to the Author**

1. Does this manuscript meet PLOS Global Public Health’s publication criteria? Is the manuscript technically sound, and do the data support the conclusions? The manuscript must describe methodologically and ethically rigorous research with conclusions that are appropriately drawn based on the data presented.

Reviewer #1: Yes

Reviewer #2: Partly

Reviewer #3: Yes

Reviewer #4: Partly

2. Has the statistical analysis been performed appropriately and rigorously?

Reviewer #1: Yes

Reviewer #2: I don't know

Reviewer #3: Yes

Reviewer #4: No

3. Have the authors made all data underlying the findings in their manuscript fully available (please refer to the Data Availability Statement at the start of the manuscript PDF file)?

Reviewer #1: Yes

Reviewer #2: Yes

Reviewer #3: Yes

Reviewer #4: No

4. Is the manuscript presented in an intelligible fashion and written in standard English?

Reviewer #1: Yes

Reviewer #2: No

Reviewer #3: Yes

Reviewer #4: Yes

5. Review Comments to the Author

Reviewer #1: Congratulations on an excellent work done.

I recommend adding a Recommendations paragraph for policy advocacy of Trauma guidelines

Perhaps think of extending the study to include more African countries or setup a Trauma registry to include more African countries.

Reviewer #2: This manuscript has some new observation and is of locoregional interest. However, there are some grammatical and syntax errors and repetition of of words in the text and references. It may be checked by an native English speaker.

The references are not uniform in style and missing dates in some. These needs to be addressed.

Reviewer #3: This manuscript describes the presentation of trauma patients to Kenyatta National Hospital. I commend the authors for collecting and presenting a large volume of important data, however, I believe the manuscript needs significant work to realise its full potential.

1. The authors give a lot of data about presentation, but there is no data on what the actual injuries were, treatments, or outcomes. Even if the focus of the study is on presentation, readers would benefit from at least brief detail on these other issues, in order to contextualise.

2. I understand that there is little epidemiological data for these patients in Kenya, but with a purely descriptive study I am left wondering what the implications of this data are? The authors have not drawn out any clear recommendations for priorities or potential solutions to improve patient care. I wonder if they might better placed to do this if they form a clear hypothesis which they can address with their data. For example, could they compare self-referred versus clinician-referred patients in terms of presentation and outcome to inform how care pathways could be improved?

3. There is a lot of repetition of information between the results text, tables, and discussion - please try to avoid repetition so that you can reduce the overall length of the manuscript to 2500 words or so. A shorter, more focussed manuscript will mean readers take away the key points you want them to have. Some tables could be moved to a supplementary file, if needed.

4. Since this manuscript is focussed on a single hospital, it would be helpful to give some context by describing the hospital, available resources, and services it offers.

6. Please include a completed STROBE checklist.

If the authors are willing to consider these changes, I would support a resubmission to this journal.

Dmitri Nepogodiev

University of Birmingham

Reviewer #4: I would like to thank you for the opportunity to review this interesting manuscript and congratulate the team on their work. Regarding the method, instead of using a full calendar year, why was the study period only 11 months? It would be very helpful to compare the results across years to arrive at a robust conclusion where data are available. It would be interesting to assess the appropriateness of these emergency visits that were treated in relation to the level of care to see if a better triage and referral system may improve the health system's performance.

Additional literature research is needed to better frame the research question, and further discussion is needed to connect the findings with health system challenges.

Lastly, the data supplied in the paper was not independent of statistical software.

6. PLOS authors have the option to publish the peer review history of their article (what does this mean?). If published, this will include your full peer review and any attached files.

**Do you want your identity to be public for this peer review?** For information about this choice, including consent withdrawal, please see our Privacy Policy.

Reviewer #1: **Yes: **Mahmoud Elfiky

Reviewer #2: No

Reviewer #3: No

Reviewer #4: No

---

## [Decision Letter · Decision Letter 1]

17 Apr 2023

Patterns of orthopedic and trauma admissions to a tertiary teaching and referral health facility in Kenya: Chart review

PGPH-D-22-01913R1

Dear Dr Omondi,

We are pleased to inform you that your manuscript 'Patterns of orthopedic and trauma admissions to a tertiary teaching and referral health facility in Kenya: Chart review' has been provisionally accepted for publication in PLOS Global Public Health.

Best regards,

Julia Robinson

Executive Editor

Reviewer Comments (if any, and for reference):

Reviewer's Responses to Questions

**Comments to the Author**

1. If the authors have adequately addressed your comments raised in a previous round of review and you feel that this manuscript is now acceptable for publication, you may indicate that here to bypass the “Comments to the Author” section, enter your conflict of interest statement in the “Confidential to Editor” section, and submit your "Accept" recommendation.

Reviewer #1: All comments have been addressed

Reviewer #2: All comments have been addressed

Reviewer #4: All comments have been addressed

2. Does this manuscript meet PLOS Global Public Health’s publication criteria? Is the manuscript technically sound, and do the data support the conclusions? The manuscript must describe methodologically and ethically rigorous research with conclusions that are appropriately drawn based on the data presented.

Reviewer #1: Yes

Reviewer #2: Yes

Reviewer #4: Yes

3. Has the statistical analysis been performed appropriately and rigorously?

Reviewer #1: Yes

Reviewer #2: I don't know

Reviewer #4: Yes

4. Have the authors made all data underlying the findings in their manuscript fully available (please refer to the Data Availability Statement at the start of the manuscript PDF file)?

Reviewer #1: Yes

Reviewer #2: Yes

Reviewer #4: Yes

5. Is the manuscript presented in an intelligible fashion and written in standard English?

Reviewer #1: Yes

Reviewer #2: Yes

Reviewer #4: Yes

6. Review Comments to the Author

Reviewer #1: Congratulations on the well prepared manuscript that shows off the hard work and dedication of the submitting team

Reviewer #2: (No Response)

Reviewer #4: (No Response)

7. PLOS authors have the option to publish the peer review history of their article (what does this mean?). If published, this will include your full peer review and any attached files.

**Do you want your identity to be public for this peer review?** For information about this choice, including consent withdrawal, please see our Privacy Policy.

Reviewer #1: **Yes: **Mahmoud Elfiky

Reviewer #2: No

Reviewer #4: No
